# Achieving stable dynamics in neural circuits

**Leo Kozachkov**[1,2,3‡], **Mikael Lundqvist**[1,2,4‡], **Jean-Jacques Slotine**[1,3‡], **Earl K. Miller**[1,2‡]*

**1** The Picower Institute for Learning & Memory, Massachusetts Institute of Technology (MIT), Cambridge, Massachusetts, United States of America, **2** Department of Brain & Cognitive Sciences, Massachusetts Institute of Technology (MIT), Cambridge, Massachusetts, United States of America, **3** Nonlinear Systems Laboratory, Massachusetts Institute of Technology (MIT), Cambridge, Massachusetts, United States of America, **4** Department of Psychology, Stockholm University, Stockholm, Sweden

‡ LK and ML share first authorship on this work. JJS and EKM are joint senior principal investigators on this work.
* ekmiller@mit.edu

**Data Availability Statement:** All detailed proofs of main results are found in the appendix. Simulations (Figs 2 and 3) were performed in Python. Code to reproduce the figures is available at [https://github.com/kozleo/stable_dynamics]. Numerical

## Abstract

The brain consists of many interconnected networks with time-varying, partially autonomous activity. There are multiple sources of noise and variation yet activity has to eventually converge to a stable, reproducible state (or sequence of states) for its computations to make sense. We approached this problem from a control-theory perspective by applying contraction analysis to recurrent neural networks. This allowed us to find mechanisms for achieving stability in multiple connected networks with biologically realistic dynamics, including synaptic plasticity and time-varying inputs. These mechanisms included inhibitory Hebbian plasticity, excitatory anti-Hebbian plasticity, synaptic sparsity and excitatory-inhibitory balance. Our findings shed light on how stable computations might be achieved despite biological complexity. Crucially, our analysis is not limited to analyzing the stability of fixed geometric objects in state space (e.g points, lines, planes), but rather the stability of state trajectories which may be complex and time-varying.

## Author summary

Stability is essential for any complex system including, and perhaps especially, the brain. The brain's neural networks are highly dynamic and noisy. Activity fluctuates from moment to moment and can be highly variable. Yet it is critical that these networks reach a consistent state (or sequence of states) for their computations to make sense. Failures in stability have consequences ranging from mild (e.g incorrect decisions) to severe (disease states). In this paper we use tools from control theory and dynamical systems theory to find mechanisms which produce stability in recurrent neural networks (RNNs). We show that a kind of "unlearning" (inhibitory Hebbian and excitatory anti-Hebbian plasticity), balance of excitation and inhibition, and sparse anatomical connectivity all lead to stability. Crucially, we focus on the stability of neural *trajectories*. This is different from traditional studies of stability of fixed points or planes. We do not assess *what* trajectories our networks will follow but, rather, when these trajectories will all converge towards each other to achieve stability.

integration was performed using sdeint, an open-source collection of numerical algorithms for performing integrations of stochastic ordinary differential equations.

**Funding:** This work was supported by NIMH R37MH087027, The MIT Picower Institute Innovation Fund, ONR MURI N00014-16-1-2832, and Swedish Research Council Starting Grant 2018-04197. The funders had no role in study design, data collection and analysis, decision to publish, or preparation of the manuscript.

**Competing interests:** The authors have declared that no competing interests exist.

## Introduction

Behavior emerges from complex neural dynamics unfolding over time in multi-area brain networks. Even in tightly controlled experimental settings, these neural dynamics often vary between identical trials [1,2]. This can be due to a variety of factors including variability in membrane potentials, inputs, plastic changes due to recent experience and so on. Yet, in spite of these fluctuations, brain networks must achieve computational stability: despite being "knocked around" by plasticity and noise, the behavioral output of the brain on two experimentally identical trials needs to be similar. How is this stability achieved?

Stability has played a central role in computational neuroscience since the 1980's, with the advent of models of associative memory that stored neural activation patterns as stable point attractors [3–7], although researchers were thinking about the brain's stability since as early as the 1950's [8]. The vast majority of this work is concerned with the stability of activity around points, lines, or planes in neural state space [9,10]. However, recent neurophysiological studies have revealed that in many cases, single-trial neural activity is highly dynamic, and therefore potentially inconsistent with a static attractor viewpoint [1,11]. Consequently, there has been a number of recent studies—both computational and experimental—which focus more broadly on the stability of neural *trajectories* [12,13], which may be complex and time-varying.

While these studies provide important empirical results and intuitions, they do not offer analytical insight into mechanisms for achieving stable trajectories in recurrent neural networks. Nor do they offer insights into achieving such stability in plastic (or multi-modal) networks. Here we focus on finding conditions that guarantee stable trajectories in recurrent neural networks and thus shed light onto how stable trajectories might be achieved *in vivo*.

To do so, we used contraction analysis, a concept developed in control theory [14]. Unlike a chaotic system where perturbations and distortions can be amplified over time, the population activity of a contracting network will converge towards the same trajectory, thus achieving stable dynamics (Fig 1). One way to understand contraction is to represent the state of a network at a given time as a point in the network's 'state-space', for instance the space spanned by the possible firing rates of all the networks' neurons. This state-space has the same number of dimensions as the number of units $n$ in the network. A particular pattern of neural firing rates corresponds to a point in this state-space. This point moves in the $n$ dimensions as the firing rates change and traces out a trajectory over time.

In a contracting network, all such trajectories converge. These contracting dynamics have previously been used in several applications, including neural networks with winner take all dynamics [15,16], in a model of action-selection in the basal ganglia [17], and to explain how neural synchronization can protect from noise [18]. Here, we instead explore how contraction can be achieved generally in more complex recurrent neural networks (RNNs) including those with plastic weights. We used RNNs that received arbitrary time-varying inputs and had synapses that changed on biologically relevant timescales [19–21]. Our analysis reveals several novel classes of mechanisms that produced contraction including inhibitory Hebbian plasticity, excitatory anti-Hebbian plasticity, excitatory-inhibitory balance, and sparse connectivity. For the first two parts of the Results section, we focus on contraction of *both* neural activity *and* components of the weight matrix (Fig 1). For the remaining parts of the Results section, we hold the weights fixed (i.e they become parameters, not variables) and focus on contraction of neural activity alone.

## Results

The main tool we used to characterize contraction was the logarithmic norm (also known as a matrix measure). The formal definition of the logarithmic norm is as follows (from [22]

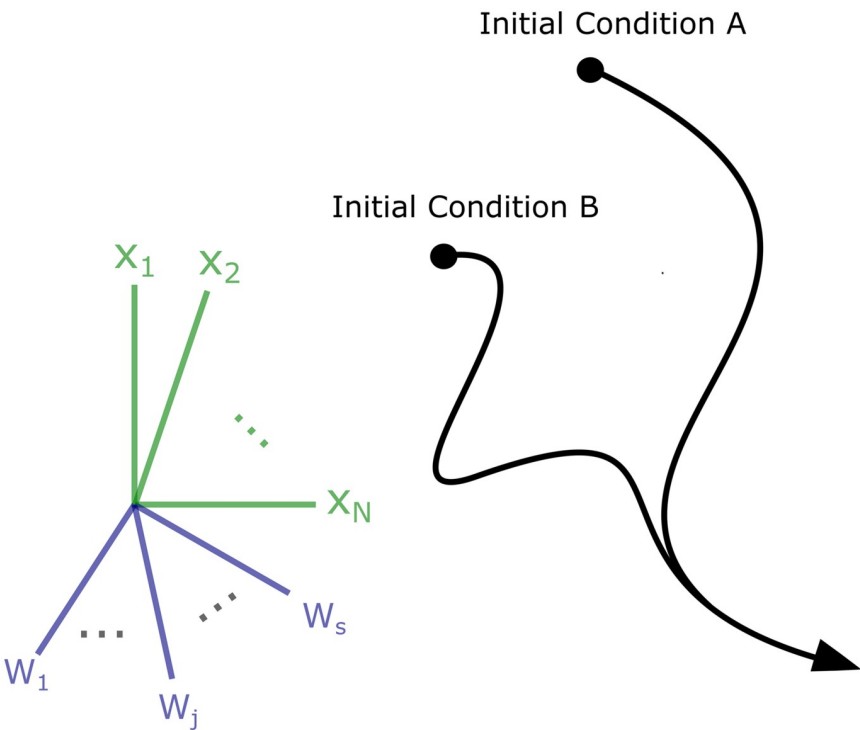

**Fig 1. Cartoon demonstrating the contraction property.** In a network with $N$ neural units and $S$ dynamic synaptic weights, the network activity can be described a trajectory over time in an $(N + S)$-dimensional space. In a contracting system all such trajectories will converge exponentially *in some metric* towards each other over time, regardless of initial conditions. In other words, the distance between any two trajectories shrinks to zero—potentially after transient divergence (as shown).

section 2.2.2): let $\mathbf{A}$ be a matrix in $C^{n \times n}$ and $\|\cdot\|_i$ be an induced matrix norm on $C^{n \times n}$. Then the corresponding logarithmic norm is the function $\mu(\cdot) : C^{n \times n} \to \mathbb{R}$ defined by

$$\mu(\mathbf{A}) = \lim_{\epsilon \to 0^+} \frac{\| I + \epsilon \mathbf{A} \|_i - 1}{\epsilon}$$

In the same way that different vector norms induce different matrix norms, different vector norms also induce different logarithmic norms. Two important logarithmic norms which we use throughout the paper are those induced by the vector 1-norm and the vector 2-norm:

$$\mu_1(\mathbf{A}) = \max_j \left[ a_{jj} + \sum_{i \neq j}^{n} |a_{ij}| \right] \qquad \mu_2(\mathbf{A}) = \lambda_{max}\left( \frac{\mathbf{A}^* + \mathbf{A}}{2} \right)$$

Where $\lambda_{max}$ denotes the largest eigenvalue. To study the contraction properties of RNNs, we applied the logarithmic norm to the RNN's *Jacobians*. The Jacobian of a dynamical system is a matrix essentially describing the local 'traffic laws' of nearby trajectories of the system in its state space. More formally, it is the matrix of partial derivatives describing how a change in any system variable impacts the *rate of change* of every other variable in the system. It was shown in [14] that if the logarithmic norm of the Jacobian is negative then all nearby trajectories are funneled towards one another (see S1A Text Section 1.2 for technical review). This, in turn, implies that *all* trajectories are funneled towards one another at rate called the *contraction rate*. The contraction rate and the logarithmic norm are related as follows: the maximum value attained by the absolute value of logarithmic norm of the Jacobian along the network's

trajectory *is* the contraction rate. In other words, if the logarithmic norm of the Jacobian is upper bounded by some negative number −*c*, where *c* > 0, then the contraction rate is simply *c*.

Importantly, the above description can be generalized to different *metrics*. A metric is a symmetric, positive definite matrix which generalizes the notion of Euclidean distance. Every invertible coordinate transformation $\mathbf{y} = \mathbf{\theta x}$ yields a metric $\mathbf{M} = \mathbf{\theta}^T\mathbf{\theta}$. To see this, consider the squared norm of $\|\mathbf{y}\|^2 = \mathbf{y}^T\mathbf{y} = \mathbf{x}^T\mathbf{\theta}^T\mathbf{\theta x} = \mathbf{x}^T\mathbf{Mx}$. Thus, the norm of $\mathbf{y}$ is related to the norm of $\mathbf{x}$ through the metric $\mathbf{M}$. If one can find metric in which the network is contracting—in the sense that its Jacobian has negative logarithmic norm–this implies contraction for *all* coordinate systems. This makes contraction analysis useful for analyzing systems where exponential convergence of trajectories is preceded by transient divergence (Fig 1) as in recent models of motor cortex [23,24]. In this case, it is usually possible to find a coordinate system in which the convergence of trajectories is 'pure'. For example, linear stable systems were recently used in the motor control literature to find initial conditions which produce the most energetic neural response [23] They are 'purely' contracting in a metric defined by the eigenvectors of the weight matrix (see Example 5.1 in [14]) but transiently diverging in the identity metric (i.e $\mathbf{M}$ = $\mathbf{I}$). Note that the identity metric corresponds to $\mathbf{\theta}$ = $\mathbf{I}$, which is simply the original, untransformed coordinate system.

## Inhibitory hebbian plasticity & excitatory anti-Hebbian plasticity produce contraction

It is known that certain forms of synaptic plasticity can quickly lead to extreme instabilities if left unchecked [9,25]. Thus, the same feature that can aid learning can also yield chaotic neural dynamics if not regulated. It is not known how the brain resolves this dilemma. A growing body of evidence—both experimental and computational—suggests that inhibitory plasticity (that is, the strengthening of inhibitory synapses) can stabilize neural dynamics while simultaneously allowing for learning/training in neural circuits [26–28]. By using the Jacobian analysis outlined above, we found that inhibitory Hebbian synaptic plasticity (as well as excitatory anti-Hebbian plasticity) indeed leads to stable dynamics in neural circuits. Specifically, we considered neural networks of the following common form:

$$\dot{x}_i = h(x_i) + \sum_{j=1}^{N} W_{ij}x_j + u_i(t) \qquad (1)$$

where the term $x_i$ denotes the 'activation' of neuron *i* as a function of time. Here we follow other authors [23] and interpret $x_i$ as the *deviation* from the baseline firing rate of neuron *i*. Note that this interpretation assumes that the baseline firing rates are positive–thus allowing for x to be negative—and large enough so that *baseline* + *x* > 0. The term $W_{ij}$ denotes the weight between neurons *i* and *j* the term $h(x_i)$ captures the dynamics neuron *i* would have in the absence of synaptic input, including self-feedback terms arising from the diagonal elements of the weight matrix—in other words, the dynamics neuron *i* would have if for all *i* and *j*, $W_{ij}$ = 0. The term being summed represents the weighted contribution of all the neurons in the network on the activity of neuron *i*. Finally, the term $u_i(t)$ represents external input into neuron *i*.

We did not constrain the inputs into the RNN (except that they were not infinite) and we did not specify the particular form of $h(x_i)$ except that it should be a leak term (i.e has a negative derivative for all *x*, see S1A Text Section 2.2.4, e.g $h(x_i)$ = −$x_i$). Furthermore, we made no assumptions regarding the relative timescales of synaptic and neural activity. Synaptic dynamics were treated on an equal footing as neural dynamics. We considered synaptic plasticity of

the following correlational form [29]:

$$\dot{W}_{ij} = -k_{ij}x_i x_j - \gamma(t)W_{ij} \qquad (2)$$

where the term $k_{ij} > 0$ is the learning rate for each synapse and $\gamma(t) > 0$ is a decay factor for each synapse. For technical reasons outlined in the appendix (S1A Text Section 3), we restricted **K**, the matrix containing the learning rates $k_{ij}$, to be positive semi-definite, symmetric, and have positive entries. A particular example of **K** satisfying these constraints is to have the learning rates of all synapses to be equal (i.e. $k_{ij} = k > 0$).

Before we show that (2) leads to overall synaptic and neural contraction, it's useful to spend some time interpreting this plasticity. Since $W_{ij}$ can be positive or negative (corresponding to excitatory and inhibitory synapses, respectively), and $x_i x_j$ can be positive or negative (corresponding to correlated and anticorrelated neurons, respectively), there are four cases to consider. We summarize these cases in Table 1 and discuss them in details below. By Hebbian plasticity we refer to the increase of synaptic efficiency between correlated neurons [30]. In the context of simple neural networks with scalar weights, as we consider here, efficiency refers to the absolute value $|w|$ of a weight. Thus, for excitatory synapses, (2) in fact describes *anti*-Hebbian plasticity, because the positive synaptic weight becomes less positive (and thus less efficient) between correlated neurons and more positive (thus more efficient) for anticorrelated neurons. For inhibitory synapses, (2) describes Hebbian plasticity because the direction of synaptic weight change is negative between correlated neurons, and thus the synapse becomes *more* efficient [31,32], while for anticorrelated neurons the direction of synaptic weight change is positive, and thus the synapse becomes less efficient. Plasticity of this form produced contracting neural and synaptic dynamics regardless of the initial values of the weights and neural activity (Figs 2 and 3). The black trace of Fig 3A shows that this is not simply due to the weights decaying to 0. Thus, this plasticity is not only contraction preserving, it is contracting *ensuring*. Furthermore, we showed that the network is contracting in a non-identity metric (which we derive from the system parameters in **K**), opening up the possibility of transient divergent dynamics in the identity metric, as seen in the modelling of motor dynamics [23].

To explain how inhibitory Hebbian plasticity and excitatory anti-Hebbian plasticity work to produce contraction across a whole network, we needed to deal with the network in a holistic fashion, not by analyzing the dynamics of single neurons. To do so, we conceptualized RNNs with dynamic synapses as a single system formed by combining two subsystems, a neural subsystem and a synaptic subsystem. We showed that the above plasticity rule led the neural and synaptic subsystems to be independently contracting. Thus contraction analysis of the overall system then boiled down to examining the interactions between these subsystems [33].

We found that this plasticity works like an interface between these systems. It produces two distinct effects that push networks toward contraction. First, it makes the synaptic weight matrix symmetric (Fig 3A, red trace). This means that the weight between neuron $i$ to $j$ is the same as $j$ to $i$. We showed this by using the fact that every matrix can be written as the sum of a

**Table 1. Summary of the effect of the plasticity described in Eq (2) on excitatory and inhibitory for correlated or anticorrelated pre and post synaptic neurons.**

|  | **Correlated Neurons** | **Anticorrelated Neurons** |
|---|---|---|
|  | $x_i x_j > 0$ | $x_i x_j < 0$ |
| **Excitatory Synapse** | Less Efficient | More Efficient |
| $w > 0$ | $\Delta|w| < 0$ | $\Delta|w| > 0$ |
| **Inhibitory Synapse** | More Efficient | Less Efficient |
| $w < 0$ | $\Delta|w| > 0$ | $\Delta|w| < 0$ |

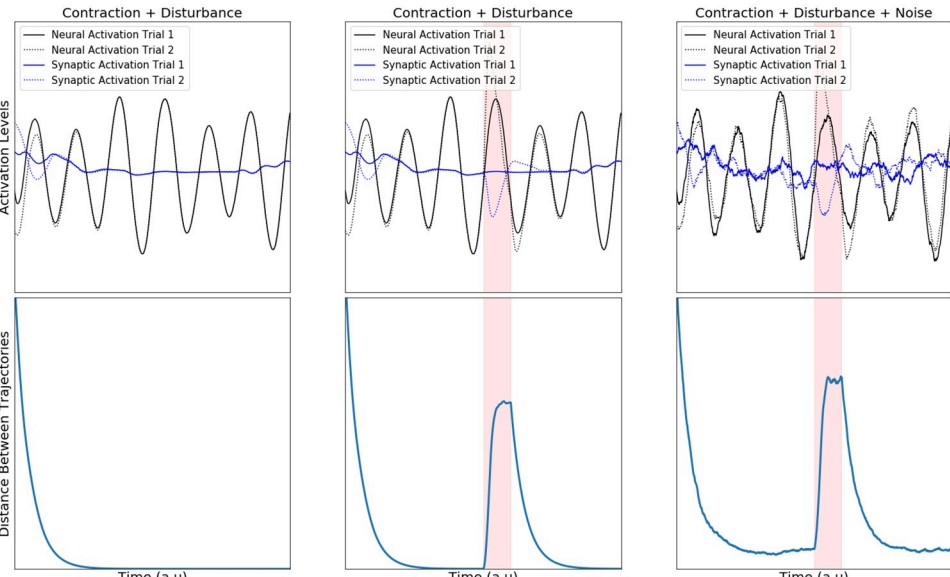

**Fig 2. Contracting dynamics of neural and synaptic activity.** Euclidean distances between synaptic and neural trajectories demonstrate exponential shrinkage over time. The top row of panels shows the activation of a randomly selected neural unit (black) and synapse (blue) across two simulations (dotted and solid line). The bottom row shows the average Euclidean distance in state space for the whole population across simulations with distinct, randomized starting conditions. Leftmost Panel: Simulations of a contracting system where only starting conditions differ over simulations. Center Panel: the same as in Leftmost but with an additional random pulse perturbation in one of the two simulations indicated by a red background shading. Rightmost Panel: same as in Center Panel but with additional sustained noise, unique to each simulation.

purely symmetric matrix and a purely anti-symmetric matrix. An anti-symmetric matrix is one where the *ij* element is the negative of the *ji* element (*i.e.* $W_{ij} = -W_{ji}$) and all the diagonal elements are zero. We then showed that anti-Hebbian plasticity shrinks the anti-symmetric part of the weight matrix to zero, implying that the weight matrix becomes symmetric. The symmetry of the weight matrix 'cancels out' off-diagonals in the Jacobian matrix (see S1A Text Section 3) of the overall neural-synaptic system. Loosely speaking, off-diagonal terms in the Jacobian represent potentially destabilizing cross-talk between the two subsystems. Furthermore, anti-Hebbian plasticity makes the weight matrix negative semi-definite. This means that all its eigenvalues are less than or equal to zero (Fig 3).

## Sparse connectivity pushes networks toward contraction

Synaptic connectivity in the brain is extraordinarily sparse. The adult human brain contains at least $10^{11}$ neurons yet each neuron forms and receives on average only $10^3$–$10^4$ synaptic connections [34]. If the brain's neurons were all-to-all connected this number would be on the order of $10^{11}$ synaptic connections per neuron ($\frac{10^{11} \cdot 10^{11}}{10^{11}} \cdot \frac{[\text{synaptic\_connections}]}{[\text{neurons}]}$). Even in local patches of cortex, such as we model here, connectivity is far from all-to-all; cortical circuits are sparse [35]. Our analyses revealed that sparse connectivity helps produce global network contraction for many types of synaptic plasticity.

To account for the possibility that some synapses may have much slower plasticity than others (and can thus be treated as synapses with fixed amplitude), we made a distinction between the total number of synapses and the total number of *plastic* synapses. These plastic synapses then changed on a similar time-scale as the neural firing rates. By neural dynamics, we mean

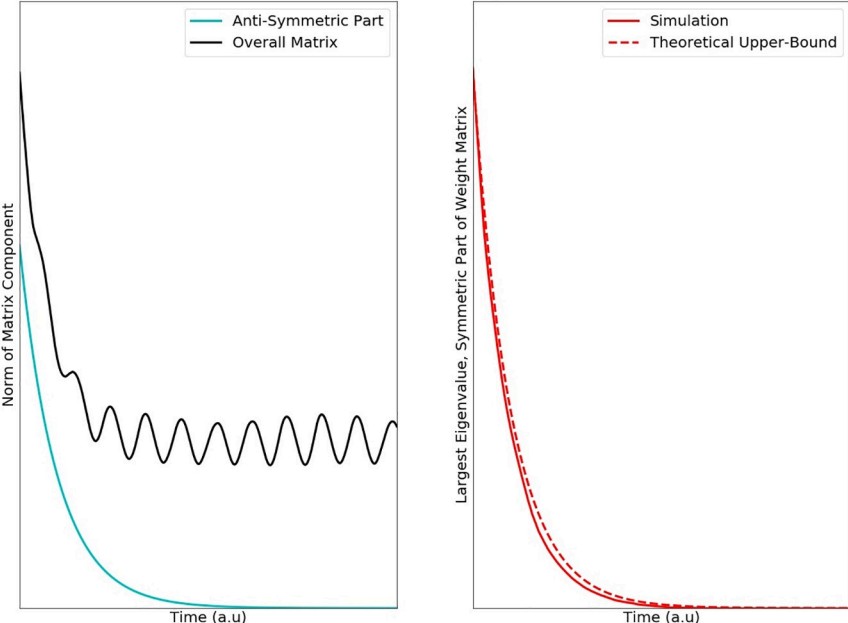

**Fig 3.** The anti-Hebbian plasticity pushes the weight matrix towards symmetry. (Left) Plotted are the spectral norms (largest singular value) of the overall weight matrix as well as the anti-symmetric part of that matrix. Since every square matrix can be uniquely decomposed as the sum of a symmetric and anti-symmetric component—0.5*(W+W') and 0.5*(W-W')', respectively—the teal curve decaying to zero implies that the matrix becomes symmetric. The black trace shows the spectral norm of the overall weight matrix. If this quantity does not decay to zero, it implies that not all the weights have decayed to zero. On the right, we plot the largest eigenvalue of the symmetric part of W. A prerequisite for overall contraction of the network is that this quantity be less than or equal to the 'leak-rate' of the individual neurons. The dotted line shows our theoretical upper bound for this quantity, and the solid line shows the actual value of taken from a simulation (see Methods).

the change in neural activity as a function of time. We analyzed RNNs with the structure:

$$\dot{x}_i = h_i(x_i) + \sum_{j=1}^{N} W_{ij}(t) \ r(x_j) + u_i(t) \tag{3}$$

Where $h_i(x_i)$ is a nonlinear leak term (see S1A Text Section 2.2.4), and $r(x_j)$ is a nonlinear activation function. The RNNs analyzed in this section are identical to those analyzed in the previous section, with the exception of the $r$ terms, which we constrained to be linear. Under the assumption that the plastic synapses have a 'forgetting term', we show in the appendix (S1A Text Section 4) that if the following equation is satisfied for every neuron, then the overall network is contracting:

$$p_i(g_{max}w_{max} + \alpha_i r_{max}) < \beta_i \tag{4}$$

where $p_i$ denotes the total number of afferent synapses into neuron $i$ and $\alpha_i$ denotes the fraction of afferent *plastic* synapses into neuron $i$. The term $w_{max}$ refers to the maximum possible absolute efficiency of any single synapse. That is, $w_{max} = \max_{i,j} |w_{ij}|$. Similarly, the term $r_{max}$ refers to the maximum possible absolute value of $r$. That is, $r_{max} = \max_{i,t} |r_i(t)|$. The term $\beta_i$ denotes the contraction rate of the $i^{th}$ isolated neuron. That is, $\beta_i = -\max_{i,t} \frac{\partial h_i}{\partial x_i}(t)$. Recall from the introduction that the contraction rate measures how quickly the trajectories of a contracting system reconvene after perturbation. Finally, $g_{max}$ refers to the maximum gain of any neuron in the network. That is, $g_{max} = \max_{i,t} |\frac{\partial r_i}{\partial x_i}|(t)$. Note that because $\beta_i$ is a positive number by

assumption, it is always possible to decrease $p_i$ to the point where (4) is satisfied. Of course, it is possible that the only value of $p_i$ that satisfies (4) is the trivial solution $p_i = 0$, which corresponds to removing all interconnections between neurons. Since these neurons are assumed to be contracting in isolation, the network is trivially contracting. However, if the term inside the parentheses of (4) is small enough, or $\beta_i$ is large enough, intermediate value of $p_i$ can be found which satisfy the inequality. Because increasing the sparsity of a network corresponds to decreasing $p_i$, we may conclude that increasing the sparsity of connections pushes the system in the direction of contraction. Note that (4) also implies that the faster the individual neurons are contracting (i.e. the larger $\beta_i$ is), the denser you can connect them with other neurons while still preserving overall contraction.

Up to now we have focused our analysis on the case where synaptic weights vary on a timescale comparable to neurons, and must therefore be factored into the stability analysis. For the next two sections, we'll apply contraction analysis to neural network in the case where the weights may be regarded as *fixed* relative to the neural dynamics (i.e. there is a separation of timescales).

## E-I balance leads to contraction in static RNNs

Apart from making connections sparse, one way to ensure contraction is to make synaptic weights small. This can be seen for the case with static synapses by setting $\alpha_i = 0$ in the section above, where $W_{max}$ now has to be small to ensure contraction. Intuitively, this is because very small weights mean that neurons cannot exert much influence on one another. If the neurons are stable before interconnection, they will remain so. Since strong synaptic weights are commonly observed in the brain, we were more interested in studying when contraction can arise irrespective of weight amplitude. Negative and positive synaptic currents are approximately balanced in biology [36–38]. We reasoned that such balance might allow much larger weight amplitudes while still preserving contraction since most of the impact of such synapses cancel and the net effect small. This was indeed the case. To show this, we studied the same RNN as in the section above, while assuming additionally that the weights are static. In particular, we show in the appendix (S1A Text Section 5) that contraction can be assessed by studying the eigenvalues of the *symmetric* part of **W** (i.e. $\frac{\mathbf{W}+\mathbf{W}^\mathrm{T}}{2}$).

Before we discuss the above result in detail, it is useful here to quickly review some facts about the stability of nonlinear systems as compared to the stability of linear systems. In particular, the fact that the eigenvalues of **W** are only informative for assessing contraction in regions where the dynamics may be regarded as linear. This is because in linear time-variant (LTI) systems (i.e. $\dot{\mathbf{x}} = \mathbf{A}\mathbf{x}$) stability is completely characterized in terms of the eigenvalues of **A**. However, this is *not* true for nonlinear systems, even those of the linear time-varying form $\dot{\mathbf{x}} = \mathbf{A}(t)\mathbf{x}$. To see this, consider the following counter-example (from [39], section 4.2.2):

$$\begin{bmatrix} \dot{x} \\ \dot{y} \end{bmatrix} = \begin{bmatrix} -1 & e^{2t} \\ 0 & -1 \end{bmatrix} \begin{bmatrix} x \\ y \end{bmatrix} \tag{5}$$

The eigenvalues of $\mathbf{A}(t)$ are $(-1, 1)$ for all time, however one can verify by direct evalution that the solution of this system satisfies $y = y(0)e^{-t}$, $\dot{x} = -x + y(0)e^t$ which is unstable along $x$. However, it can be shown straightforwardly that if the eigenvalues of the *symmetric* part of $\mathbf{A}(t)$ are all negative, then the system is stable [39]. This fact underlies our analysis, and highlights the reason why the eigenvalues of the symmetric part of **W** are important for stability.

Returning to our results, we show that if excitatory to inhibitory connections are of equal amplitude (and opposite sign) as inhibitory to excitatory connections, they will not interfere negatively with stability—regardless of amplitude (see S1A Text Section 5). This is because connections between inhibitory and excitatory units will be in the off-diagonal of the overall weight matrix and get cancelled out when computing the symmetric part. As an intuitive example, consider a two-neuron circuit made of one excitatory neuron and one inhibitory neuron connected recurrently (as in [40], Fig 1A). Assume that the overall weight matrix has the following structure:

$$\mathbf{W} = \begin{pmatrix} w & -w \\ w & -w \end{pmatrix}$$

When taking the symmetric part of this matrix, the off-diagonal elements cancel out, leaving only the diagonal elements to consider. Since the eigenvalues of a diagonal matrix are simply its diagonal elements, we can conclude that if the excitatory and inhibitory subpopulations are independently contracting ($w$ is less than the contraction rate of an isolated neuron), then overall contraction is guaranteed. It is straightforward to generalize this simple two-neuron example to circuits achieving E-I balance through interacting *populations* (see S1A Text Section 5). It is also straightforward to generalize to the case where E-I and I-E connections do not cancel out exactly neuron by neuron, but rather they cancel out in a statistical sense where the mean amplitudes are matched. Another way to view this E-I balance is in the framework of combinations of contracting systems (Fig 4). It is known that combining independently contracting systems in negative feedback preserves contraction [14]. We show that E-I balance actually translates to this negative feedback and thus can preserve contraction.

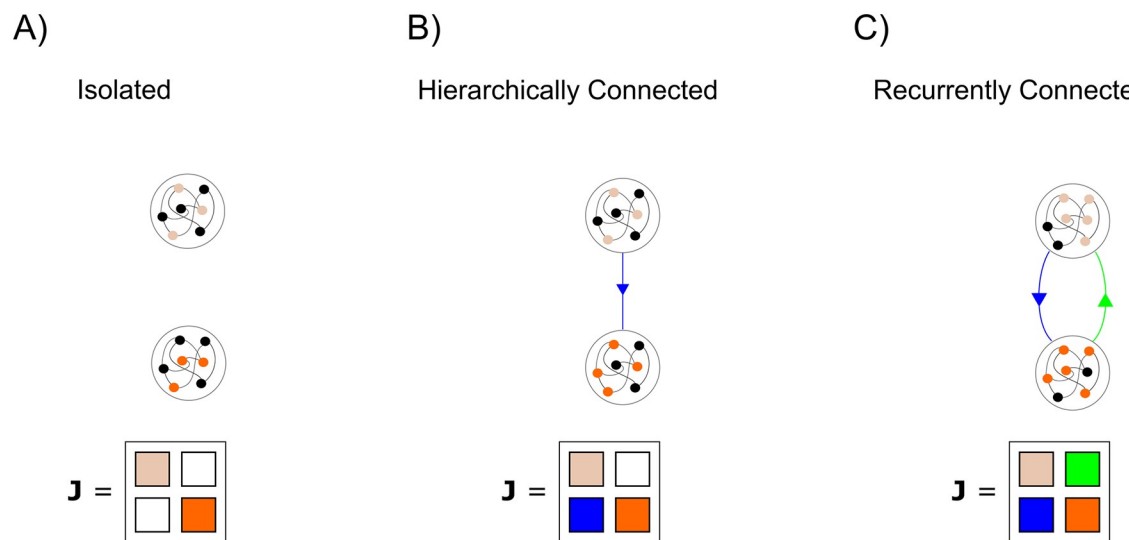

**Fig 4. Cartoon illustrating the combination properties of contracting systems.** A) Two isolated, contracting systems. The Jacobian of the overall system is block diagonal, with all zeros on the off-diagonal—corresponding to the fact that the systems are not connected. B) If one of the systems is connected to the other in a feedforward manner, the overall Jacobian is changed by the presence of non-zero terms on the bottom left block—corresponding to the connections going from the 'top' system to the 'bottom' system. This Jacobian may not be negative definite. However, it is known that a coordinate change exists which will make it negative definite. Thus, hierarchically connected contracting systems are contracting. C) If the systems are reciprocally connected, the system may lose its contracting properties (for example in the case of positive feedback). However, it is known that if the feedforward connections (blue) are 'equal and opposite' to the feedback connections (green) then the overall system is contracting. We use this property in the main text to prove that inhibitory Hebbian plasticity and excitatory anti-Hebbian plasticity lead to contracting neural circuits.

### Relation to other models with fading memory

As can be seen in Fig 2, contracting systems have 'fading memories'. This means that past events will affect the current state, but that the impact of a transient perturbation gradually decays over time. Consider the transient input in Fig 2 (red panel) presented on only one of the two trials to the network. Because the input is only present on one trial and not the other we call it a perturbation. When this perturbation occurs, the trajectories of the two trials become separated. However, after the disturbance is removed, the distance between the network's trajectories starts shrinking back to zero again. Thus, the network does not hold onto the memory of the perturbation indefinitely—the memory fades away. A similar property has been used in Echo State Networks (ESNs) and liquid state machines (LSMs) to perform useful brain-inspired computations [41,42]. These networks are an alternative to classical attractor models in which neural computations are performed by entering stable states rather than by 'fading memories' of external inputs [43].

While there are several distinctions between the networks described above and ESNs (e.g. ESNs are typically discrete time dynamical systems, rather than continuous), we show in the appendix (S1A Text Section 5.1) that they are a special case of the networks considered here. We show this for ESNs as opposed to LSMs because LSMs are typically implemented on integrate and fire neurons which, because of the spike reset, have a sharp discontinuity in their dynamics—making them unamenable to contraction analysis.

By highlighting the link between contraction and ESNs, we demonstrate that the contracting neural networks considered here are in principle capable of performing useful and interesting neural computations. In other words, the strong stability properties of contracting neural networks do not automatically prohibit them from doing interesting computations. By working within the framework of contraction analysis we were able to study networks both with dynamic synapses and non-identity metrics—a much broader model space than allowed by the standard ESN framework.

## Discussion

We studied a fundamental question in neuroscience: How do neural circuits maintain stable dynamics in the presence of disturbance, noisy inputs and plastic change? We approached this problem from the perspective of dynamical systems theory, in light of the recent successes of understand neural circuits as dynamical systems [44]. We focused on *contracting* dynamical systems, which are yet largely unexplored in neuroscience, as a solution to the problem outlined above. We did so for three reasons:

1. Contracting systems can be *input-driven*. This is important because neural circuits are typically bombarded with time-varying inputs either from the environment or from other brain areas. Previous stability analyses have focused primarily on the stability of RNNs without time-varying input. These analyses are most insightful in situations where the input into a circuit can be approximated as either absent or constant. However, naturalistic stimuli tend to be highly time-varying and complex [45].

2. Contracting systems are robust to noise and disturbances. Perturbations to a contracting system are forgotten at the rate of the contraction and noise therefore does not stack up over time. Importantly, the rate of forgetting (i.e the contraction rate) does not change with the size of the perturbation. Thus dynamic stability can co-exist with high trial-to-trial variability in contracting neural networks, as observed in biology.

3. Contracting systems can be combined with one another in ways that preserve contraction (Fig 4). This is not true of most dynamical systems which can easily 'blow up' when connected in feedback with one another [8]. This combination property is important as it is increasingly clear that cognitive functions such as working memory or attention are distributed in multiple cortical and sub-cortical regions [46,47]. In particular, prefrontal cortex has been suggested as a hub that can reconfigure the cortical effective network based on task demands [48]. Brain networks must therefore be able to effectively reconfigure themselves on a fast time-scale without loss of stability. Most attempts in modelling cognition, for instance working memory, tend to utilize single and often autonomous networks. Contracting networks display a combination of input-driven and autonomous dynamics, and thus have key features necessary for combining modules into flexible and distributed networks.

To understand what mechanisms lead to contraction in neural circuits, we applied contraction analysis to RNNs. For RNNs with static weights, we found that the well- known Echo State Networks are a special case of a contracting network. Since realistic synapses are complex dynamical systems in their own right, we went one step further and asked when neural circuits with dynamic synapses would be contracting. We found that inhibitory Hebbian plasticity as well as excitatory anti-Hebbian plasticity and synaptic sparsity all lead to contraction in a broad class of RNNs.

Inhibitory plasticity has recently been the focus of many experimental and computational studies due to its stabilizing nature as well as its capacity for facilitating nontrivial computations in neural circuits [27,28,49]. It is known to give rise to excitatory-inhibitory balance and has been implicated as the mechanism behind many experimental findings such as sparse firing rates in cortex [28]. Similarly, anti-Hebbian plasticity exists across many brain areas and species, such as salamander and rabbit retina [31], rat hippocampus [50,51], electric fish electrosensory lobe [52] and mouse prefrontal cortex [53]. Anti-Hebbian dynamics can give rise to sparse neural codes which decrease correlations between neural activity and increase overall stimulus representation in the network [54]. Because of this on-line decorrelation property, anti-Hebbian plasticity has also been implicated in predictive coding [31,52]. Our findings suggest that it also increase the stability of networks.

For more general forms of synaptic dynamics, we showed that synaptic sparsity pushes RNNs towards being contracting. This aligns well with the experimental observation that synaptic connectivity is typically extremely sparse in the brain. Our results suggest that sparsity may be one factor pushing the brain towards dynamical stability. It is therefore interesting that synapses are regulated by homeostatic processes where synapses neighboring an upregulated synapse are immediately downregulated [55]. On the same note, we also observed that balancing the connections between excitatory and inhibitory populations leads to contraction. Balance between excitatory and inhibitory synaptic inputs are often observed in biology [36–38], and could thus serve contractive stability purposes. Related computational work on spiking networks has suggested that balanced synaptic currents leads to fast response properties, efficient coding, increased robustness of function and can support complex dynamics related to movements [21,56–58].

A main advantage to our approach is that it provides provable certificates of global contractive stability for nonlinear, time-varying RNNs with synaptic plasticity. This distinguishes it from previous works where—while very interesting and useful—stability is experimentally observed, but not proven [12]. In some cases [23,24], linear stability around the origin is proven (which implies that there is a contraction region around the origin) but the size of this region is neither established nor sought after. Indeed, one future direction we are pursuing is

the question of: *given an RNN, can one provide a certificate of contractive stability in a region*? An answer to this question would shed light on the stability properties of known RNN models in the literature (e.g. trained RNNs, biologically-detailed spiking models, etc.).

Experimental neuroscience is moving in the direction of studying many interacting neural circuits simultaneously. This is fueled by the expanding capabilities of recording multiple areas simultaneously in vivo and study their interactions. This increases the need for multi-modal cognitive models. We therefore anticipate that the presented work can provide a useful foundation for how cognition in noisy and distributed computational networks can be understood.

## Materials and methods

In the interested of space and cohesion, we've placed all the detailed proofs of main results into the appendix. The appendix was written to be self-contained, and thus also contains additional definitions of mathematical objects used throughout the text. Simulations (Figs 2 and 3) were performed in Python. Code to reproduce the figures is available at [https://github.com/kozleo/stable_dynamics]. Numerical integrating was performed using sdeint, an open-source collection of numerical algorithms for integrating stochastic ordinary differential equations.

Fig 2 **details**:

All parameters and time constants in Eqs (1) and (2) were set to one. The integration step-size, $dt$, was set to 1e-2.

Initial conditions for both neural and synaptic activation were drawn uniformly between -1 and 1. Inputs into the network were generated by drawing $N$ frequencies uniformly between $dt$ and $100dt$, phases between 0 and $2\pi$, amplitudes between 0 and 20 and generating an N x *Time* vector of sinusoids with the above parameters.

The perturbations of the network was achieved by adding a vector of all 10s (i.e an additive vector input into the network, with each network of the element equal to 10) to the above input on one of the trials for 100 time steps in the middle of the simulation.

The noise was generated by driving each neural unit with an independent Weiner process (sigma = .2).

Fig 3 **details**:

The weight matrix used was the same as in Fig 2, leftmost panel (without perturbation, without noise).

## Supporting information

**S1 Text. The supplementary appendix file contains extensive mathematical proofs of the results stated above.** We kept the appendix self-contained by restating the basic results of contraction analysis and linear algebra which we used often in our proofs.
(PDF)

## Acknowledgments

We thank Pawel Herman for comments on an earlier version of this manuscript. We thank Michael Happ and all members of the Miller Lab for helpful discussions and suggestions.

## Author Contributions

**Conceptualization:** Leo Kozachkov, Mikael Lundqvist, Jean-Jacques Slotine, Earl K. Miller.

**Formal analysis:** Leo Kozachkov, Mikael Lundqvist.

**Project administration:** Earl K. Miller.

**Software:** Leo Kozachkov, Mikael Lundqvist.

**Supervision:** Jean-Jacques Slotine, Earl K. Miller.

**Validation:** Jean-Jacques Slotine, Earl K. Miller.

**Writing – original draft:** Leo Kozachkov, Earl K. Miller.

**Writing – review & editing:** Jean-Jacques Slotine, Earl K. Miller.

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
