## [Decision Letter · Decision Letter 0]

21 Apr 2020

Dear Dr. Miller,

Thank you very much for submitting your manuscript "Achieving stable dynamics in neural circuits" for consideration at PLOS Computational Biology.

As with all papers reviewed by the journal, your manuscript was reviewed by members of the editorial board and by two independent reviewers. We apologize for the lengthy review process for this manuscript which was due to one reviewer being unable to complete their review.

The reviewers appreciated the novel perspective on an important problem, but did raise a number of concerns and felt that several parts of the paper required further clarification. In light of the reviews (below this email), we would like to invite the resubmission of a significantly-revised version that takes into account the reviewers' comments.

We cannot make any decision about publication until we have seen the revised manuscript and your response to the reviewers' comments. Your revised manuscript is also likely to be sent to reviewers for further evaluation.

Sincerely,

Adrian M Haith

Associate Editor

PLOS Computational Biology

Lyle Graham

Deputy Editor

PLOS Computational Biology

Reviewer's Responses to Questions

**Comments to the Authors:**

Reviewer #1: The authors have some interesting results that would be useful for theorists/modelers in computational neuroscience to be aware of. The usefulness would be enhanced if some extra work is done to clarify the findings and state the limits on their generality or whether they could be expanded. I point out the areas where this can be achieved below:

l.57 “all such trajectories converge”: this suggests the focus is on networks with a single fixed point. Are these really the interesting ones? Is there some generalization to networks with multiple fixed points where one can discuss regions of contraction? Such networks may be more versatile and useful.

Eq. 1: The description “in the absence of input from other neurons” suggests that the diagonal part of the connection matrix, W_ij is contained within the first term, h(x). Is this true? Or are you only considering W_ij without diagonal self-connection terms? If not, then “other” neurons should be modified to include self-feedback.

Eq. 1: It is important to indicate the range of “x” as it appears to enter the expression as if it is a firing rate which can only be non-negative. Are you allowing negative neural outputs?

Eq.2: Following the above comment, if “x” represents firing rate so can only be non-negative, then the equation for changes in connection strengths prevent any stable positive connections from arising (the initial term is negative or zero and the second term has opposite sign to W). This seems wrong. Similarly if x can be negative some discussion should be made about the meaning of negative rates and how the change in synaptic strength should depend on such rates. Perhaps zero “x” corresponds to a threshold for the rate-dependent switch between synaptic depression and potentiation and the formulae would all work with shifted “x” and threshold subtracted, to represent non-negative firing rates?

l. 130: I think the logic is correct but the explanation a little too brief:

According to Hebb’s rule if a neuron helps cause another neuron to fire then a change in connection enhances the likelihood of such a causal event occurring again. i.e. Hebb’s rule is all about positive feedback (the synapse changes to enhance existing correlations).

For excitatory neurons that means strengthening following correlated spikes.

For an inhibitory neuron, it can help a downstream neuron fire by not spiking at the same time.

Also, a decrease in the inhibitory synapse will help the postsynaptic neuron fire.

So, firing at different times (anticorrelated) should lead to a decrease in inhibitory strength (Wij more positive) as it is then more likely that the inhibitory cell helps the excitatory cell fire -- or conversely, firing at the same time leads to an increase in inhibitory strength (Wij more negative as stated here). Perhaps a short-hand could be that a “positive feedback” process is Hebbian and a negative feedback process is anti-Hebbian? However, I am still confused by the values of “x” and their sign being perhaps negative. It may help to describe this as negative x meaning low-rate and positive x meaning high-rate. But then this rule would allow for strong plasticity when both pre- and post-synaptic rates are minimal (x very negative) in contradiction to known plasticity mechanisms. These issues should be discussed.

Figure 3: Please define “Norm of Matrix Component” via an equation

l. 146 typo (“of”)

l. 173-174 The calculation is incorrect. You are comparing connections per neuron with total connections in the brain. The numerator should be multiplied by number of neurons (10^11).

l. 197: It seems that there are two separate (but linked) systems operating on very different time scales. On the fast time scale, for a given weight matrix, one can ask if the neural firing rates of the system are contracting. At times (especially when Echo State networks are compared or the requirements for weight matrices) it seems like the goal is to generate contracting networks in terms of neural firing rates. However, l. 197 suggests that the discussions of contraction are in the components of the weight matrix. The authors should clearly separate discussions and results concerning weight matrices that contract (such that the network produces a reliable/reproducible – perhaps in this case temporally variable – set of synaptic strengths) versus whether the firing rates of neurons follow relatively stable trajectories.

Eq. 4: The terms here need a little more description and explanation of how they enter. While the main point is trivial (decreasing p can lead to stability) the terms like “g_max” appear to be related to the maximum derivative of connection strength with respect to neural activation (Eq. 4.0.3) rather than what is generally termed “gain” of a neuron (gradient of f-I curve). Also, does w_max refer to maximum strength of plastic synapses (in which case why is it not in the “alpha” term) or is it simply the maximum value of the assigned fixed synapses? Perhaps explain why increasing the fraction of static to plastic synapses allows for a larger w_max.

l. 211: “has” not “have”

l. 218 “impact of such synapses cancel” but there are time-differences in practice that commonly lead to oscillations. I think “cancel” is an overstatement.

l.224-225 “will not interfere with stability”: in fact the antisymmetric part can enhance stability (this is easy to see from the determinant of a 2x2 matrix). So, while the intent is correct, the point could be made a little more clearly. That is, if the symmetric part is stable the complete matrix will be, however, if the symmetric part is not stable the antisymmetric part could be sufficient to generate stability overall.

l. 232 type “that” should be “the”

l. 232 “the off-diagonal elements cancel out”. No! They impact the determinant of the matrix and hence can impact the values (and signs) of the eigenvalues, therefore impacting stability. But again, I think only in the sense of enhancing stability.

Fig. 4: legend is pretty opaque and stops half-way through a sentence in my version.

Section 3.4 on Echo State networks: this seems a little strange and overly expounded, especially when discussing the discrete-time nature of Echo State networks when there are almost identical Liquid State networks that gained prominence at the same time (sometimes in joint reviews) with the major difference being the Liquid State networks operate in continuous time. Why focus on Echo State not Liquid State given the latter is more similar to the current paper?

Discussion: I think papers that train some chaotic networks, for example Laje and Buonomano 2012, should be mentioned at a minimum and, even better, discussed in terms of contracting trajectories given that is what they appear to do. How is their plasticity rule or network similar or dissimilar to any mentioned in this paper that lead to contraction? Or is there system not contracting? Or only contracting in a portion of phase space? It would be nice to see this sort of analysis (even if only descriptive) applied to a few prior networks that appear to achieve contraction

l. 346 “increases” not “increase”

Reviewer #2: Overview: The authors analyze the mechanisms that neural circuits employ in order to maintain stable dynamics in face of perturbations and plasticity through the lens of a control theoretic perspective, namely, contraction analysis. The study indicates that mechanisms such as inhibitory Hebbian plasticity, excitatory anti-Hebbian plasticity, synaptic sparsity and E-I balance can all lead to stability within the considered networks.

Comments:

• Contribution: The authors point out that stability of neural circuits has been considered in the literature widely, but what is less explored is how despite trial to trial variability, the brain manages to produce robust computations and therefore robust responses. The method considered here i.e., contraction analysis, states that the trajectories produced by population activity converge quickly following perturbation and therefore are robust in the face of small perturbations.

The paper brings together certain established analytical methods (contraction theory) to treat a long standing problem in computational neuroscience. While some of the factors discussed (inhibitory plasticity, E-I balance) have been studied in the context of achieving stability, they are brought together here under a common analytical framework. In this sense, the paper could constitute a useful contribution to the literature. However, the technical novelty is at time ambiguous and the presentation could be more precise in this regard. I elaborate on specific concerns below:

• Main concerns.

o The authors frequently refer to computations in the Appendix section to narrate their findings. Important quantifications (not the extensive mathematical derivations) should be introduced in line with the claims made in the Result section to make the narrative more cohesive. For instance, in line 80 the authors introduce a matrix measure called the ‘logarithmic norm of the Jacobian’ which happens to be an important metric for the discussions that follow, but no concise mathematical form of it is provided to aid the readers.

o The authors have identified two quantitative tools to characterize contraction: contraction rate and the Jacobian of the network. However, in line 85 the authors have expressed one in terms of the other. Does that mean that the quantifications are interchangeable, and the analysis can be interpreted in terms of any one? Some explicit clarifications regarding this might be helpful.

o In line 86, the authors state that the idea of contraction analysis using Jacobian can be generalized to different metrics. What are some of these metrics? What are the metrics that can generate pure convergence of trajectories? What is the identity metric in line 138? A discussion along these lines including the mathematical expression of these metrics will enrich the study.

o The types of perturbations considered for the study are not mentioned explicitly. Does the nature of convergence change depending upon the perturbation inflicted on the network?

o In line 211 the authors state that if alpha_i = 0, then W_max must be reduced to ensure equation 4 i.e., contraction in the network. My question here is in this case do you consider a densely connected network? Since, equation 4 states that if you are sparse enough you can have strong couplings irrespective of whether the synapses are static or dynamic.

o How should Figure 4 in the manuscript be interpreted? I understand that it points to the fact that E-I balance in networks (often employed by achieving E-I balance in subnetworks) facilitate stability and therefore convergence of trajectories under perturbation. But much of the preceding and succeeding discussion assumes prior knowledge of the matter. Introducing some key equations back into the text here will help justify the claims made.

o Are there any neural mechanisms that have been studied experimentally to ensure stability and robustness in computation but cannot be explained solely through contraction analysis?

• Minor Comments:

o Figure 1: Dimensionality of the cartoon state space. The text states that it is a (N+S) dimensional space for a network with N neural units and S dynamic synaptic weights, but the figure does not match up with it or perhaps need to be restructured. What is the index j here?

o Figure 2: Legends are not visible. Consider using larger font size.

o Line 157: There is no red trace in Figure 3A. Is it a typographical error or a missing trace?

o Caption for Figure 4 is incomplete.

o The Materials and Methods section does not add much in this version of the manuscript except where to find the relevant code. Consider expanding this section to include information that are vital to interpretation of the results.

o What is H in the supplementary section (Pg 13). Is it simply H = x_t x_t^T?

**Have all data underlying the figures and results presented in the manuscript been provided?**

Reviewer #1: Yes

Reviewer #2: No: Simulations are specified, but data is not submitted.

PLOS authors have the option to publish the peer review history of their article (what does this mean?). If published, this will include your full peer review and any attached files.

Reviewer #1: No

Reviewer #2: No
---

## [Decision Letter · Decision Letter 1]

27 Jun 2020

Dear Dr. Miller,

We are pleased to inform you that your manuscript 'Achieving stable dynamics in neural circuits' has been provisionally accepted for publication in PLOS Computational Biology.

Best regards,

Adrian M Haith

Associate Editor

PLOS Computational Biology

Lyle Graham

Deputy Editor

PLOS Computational Biology

Reviewer's Responses to Questions

**Comments to the Authors:**

Reviewer #1: All prior criticisms have been addressed satisfactorily and the paper is clear enough now.

Please just carefully check the grammar before sending it for any publication as there are some typos/non-English forms in the added text (e.g. l.120 "at rate called the contraction rate" should begin "at a rate"). I came across a few of these sorts of issues.

Reviewer #2: The authors have addressed the points raised in my original review. I thank them for their responses and am satisfied with the edits made to the manuscript.

**Have all data underlying the figures and results presented in the manuscript been provided?**

Reviewer #1: Yes

Reviewer #2: None

PLOS authors have the option to publish the peer review history of their article (what does this mean?). If published, this will include your full peer review and any attached files.

Reviewer #1: **Yes: **Paul Miller

Reviewer #2: No

---

## [Editor Report · Acceptance letter]

30 Jul 2020

PCOMPBIOL-D-20-00042R1 

Achieving stable dynamics in neural circuits

Dear Dr Miller,

I am pleased to inform you that your manuscript has been formally accepted for publication in PLOS Computational Biology. Your manuscript is now with our production department and you will be notified of the publication date in due course.

With kind regards,

Laura Mallard
